# Citizen Science for Marine Litter Detection and Classification on Unmanned Aerial Vehicle Images

**Silvia Merlino** [1,*] **, Marco Paterni** [2] **, Marina Locritani** [3] **, Umberto Andriolo** [4] **, Gil Gonçalves** [4,5] **and Luciano Massetti** [6]

1. Istituto di Scienze Marine del Consiglio Nazionale delle Ricerche (ISMAR-CNR), 19032 Lerici, Italy
2. Istituto di Fisiologia Clinica del Consiglio Nazionale delle Ricerche (IFC-CNR), 56124 Pisa, Italy; marco.paterni@ifc.cnr.it
3. Istituto Nazionale di Geofisica e Vulcanologia, 00143 Rome, Italy; marina.locritani@ingv.it
4. INESC Coimbra, Department of Electrical and Computer Engineering, Polo 2, 3030-290 Coimbra, Portugal; uandriolo@mat.uc.pt (U.A.); gil@mat.uc.pt (G.G.)
5. Department of Mathematics, University of Coimbra, Polo 1, Apartado 3008, 3001-501 Coimbra, Portugal
6. Istituto per la BioEconomia del Consiglio Nazionale delle Ricerche (IBE-CNR), 50019 Sesto Fiorentino, Italy; luciano.massetti@ibe.cnr.it
* Correspondence: silvia.merlino@sp.ismar.cnr.it

**Abstract:** Unmanned aerial vehicles (UAV, aka drones) are being used for mapping macro-litter in the environment. As drone images require a manual processing task for detecting marine litter, it is of interest to evaluate the accuracy of non-expert citizen science operators (CSO) in performing this task. Students from Italian secondary schools (in this work, the CSO) were invited to identify, mark, and classify stranded litter items on a UAV orthophoto collected on an Italian beach. A specific training program and working tools were developed for the aim. The comparison with the standard in situ visual census survey returned a general underestimation (50%) of items. However, marine litter bulk categorisation was fairly in agreement with the in situ survey, especially for sources classification. The concordance level among CSO ranged between 60% and 91%, depending on the item properties considered (type, material, and colour). As the assessment accuracy was in line with previous works developed by experts, remote detection of marine litter on UAV images can be improved through citizen science programs, upon an appropriate training plan and provision of specific tools.

**Keywords:** plastic; remote sensing; waste management; coastal pollution; beach; drone

## 1. Introduction

Unmanned aerial vehicles (UAV, aka drones) are being used for monitoring macro marine litter (>2.5 mm, [1]) (hereafter, ML) in various marine environments such as beaches [2–8], coastal dunes [9,10], lakeshores [11], remote islands [12], sea surface [13–15], and river waters [16].

On coasts, compared to traditional and standardized manual census surveys (e.g., [1,17]), the use of UAVs requires much less human effort in the field and thus can potentially increase the survey frequency. Moreover, UAV-based surveys are not intrusive and reduce the anthropogenic impact on the coast, an important aspect especially for sensitive areas such as dunes [9,10] and marine-protected areas [8]. Finally, as UAV images allow the geo-localisation of ML, it is possible to identify recurrent hotspots to improve knowledge of ML accumulation processes on coasts [5,8,9]. These assessments can serve, for instance, to optimize ML dynamic models (e.g., [18–20]) and stranded debris management [21,22].

The limitations of UAV-based surveys, when compared to the traditional census, reside in the less-detailed identification of ML. The survey can be based on the manual [4,8,10] or automated [2,23–33] image processing of UAV acquisitions. Manual image screening (hereafter, MS) consists in visually analysing UAV images (or orthophoto) and marking

ML, generally in a GIS environment. Automatic ML detection would be preferable to the manual procedure, as it is less tedious and demands less human effort; however, current proposed automated methodologies still lack the ability to categorising ML items in a detailed manner [27]. On the other hand, MS is highly subjective; thus, the quality of the assessment may depend on several factors such as operator experience and expertise, among others.

The consistency of MS was previously investigated by Andriolo et al. [34], who evaluated the different assessment by a group of expert operators. Results pointed out that the number of items marked on images depended more on the knowledge of common items found on the site, and thus on the territoriality, than on the expertise of the operators. This suggests that the UAV-based litter abundance map could be produced by briefly trained personnel, such as operators recruited from emerging citizen science projects.

The use of citizen science can be a potential and valid help in the UAV-based litter survey and MS performance. In recent years, researchers have been supported by volunteers and students in collecting data on beached marine litter [35–42]; therefore, the implementation of citizen science projects in schools could take advantage of synergies between educational and research goals. Training workshops are used to facilitate citizen science in classrooms and improve scientific literacy for participants [43]. Moreover, the use of advanced software tools and data (e.g., GIS) can provide new skills to participants, useful for their future career and social innovation [44].

The present work presents a citizen science program targeting secondary school students (between the age of 16 and 18) for performing the MS on aerial photos taken by a drone. The program was designed for an online working context (imposed during the COVID-19 pandemic). A training course was held for briefing the students, who were also provided with information material and a personalised QGIS application (Development Team, 2020 QGIS Geographic Information System. Open Source Geospatial Foundation Project. http://qgis.osgeo.org, accessed on 16 June 2021) for the marking and classification of ML items.

The work aimed at (i) assessing the inherent variability of MS when performed by different operators, (ii) evaluating the quality of the MS output from citizen science program, and (iii) suggesting future operational improvements for the MS optimisations.

## 2. Materials and Methods

### 2.1. Study Site and Image Dataset

The study area (Figure 1) was a sandy beach located downdrift from the Arno river estuary, within the marine protected area (MPA) of Migliarino, Massacciuccoli, and San Rossore park (SRPRK). The whole beach extends for about 11 km along shore, with an N–S orientation, limited southwards by a 150 m long semi-submerged groin and backward by a dune system reaching a maximum height of about 7 m [45]. The tidal regime is micro-tidal, the wave climate is characterised by a dominant southwesterly wave direction, with wave heights of usually about 1 m [46].

This coastal stretch, located between the two rivers Arno (N 4340′47.408′′, E 1016′40.466′′) and Serchio (N 4347′1.704′′, E 1016′0.016′′), is affected by a long-shore current that goes from the mouth of Arno northward, with a considerable transport of fluvial material. The Arno River is an important Italian watercourse that crosses the Tuscany region, running through large cities such as Florence and Pisa and industrial and production centers such as the province of Prato and Pontedera. The coast is also subjected to important coastal erosion phenomena [47,48], which influence the dynamics of the accumulation of sediments and materials transported by the river [47].

The selected study area, called Test Area (Figure 1), was a 900 m$^2$ portion of San Rossore beach, situated between the swash zone and the upper beach dune toe. Access to the study area is forbidden for recreational purposes and only allowed for research activities upon permission.

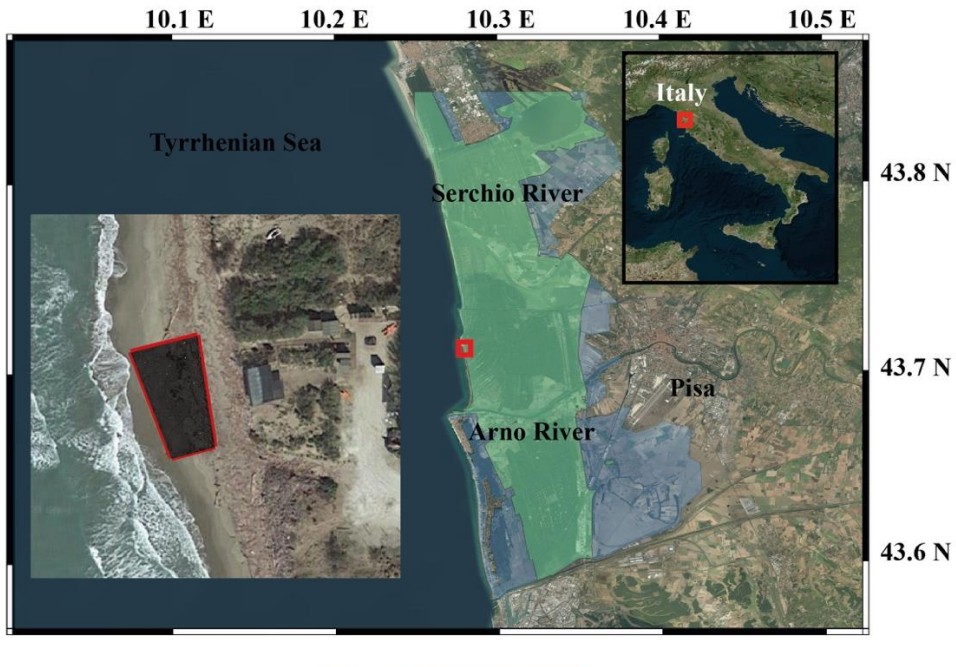

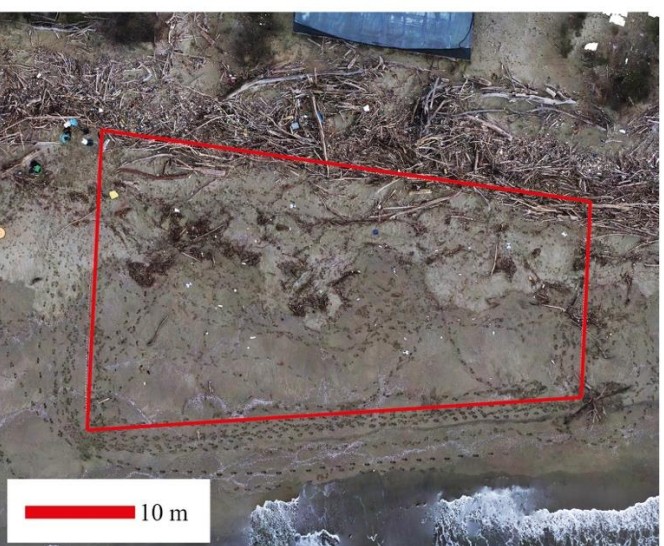

**Figure 1.** Study area location. Map of Migliarino, Massacciuccoli, and San Rossore park (upper green area) on the Tuscanian coast and satellite image of the study site (inset). Below, aerial image acquired by drone (flight height 15 m) of the target area (red trapezoidal area).

A multirotor DJI Phantom 4 Pro v2 quadcopter, with the camera (1 inch, 20 megapixel CMOS sensor, 24 mm full-frame equivalent) gimbal set to 90 degrees looking at the nadir (perpendicular to flight direction), acquired high-resolution images flying at 15 m height. Images were recorded with 80% front and 80% side overlaps.

The UAV operated automatically using the Drone Harmony (DH) ground station software. Following previous similar studies [8,34], we chose a flight height of 15 m as the right compromise between ground sampling distance (GSD) and area coverage. This setting allowed us to fly over the Test Area in 10 min.

From the image dataset, the digital surface model (DSM) and the orthophoto beach map were produced, applying a Structure from Motion-MultiView Stereo (SfM-MVS) photogrammetric processing on Agisoft Metashape [49–52]. The final orthophoto GSD resolution was 0.41cm/pixel (Figure 1).

*2.2. Manual Image Screening and Training Framework*

For performing the manual image screening (MS) and marking ML on drone images, similarly to Merlino et al. [8], the image was tiled with a 3 × 3 m square grid to make the MS regular and organised. An ad hoc user-friendly QGIS application interface, based on a drop-down list of tasks, was developed to guide the operator. After the recognition of an ML item on image, the operator was asked to digitise the item contour using the graphical existing tool available in QGIS. The software automatically retrieved the geometric properties of the object (area, length, and GPS coordinates of the centroid), discarding objects smaller than 5 cm. In the second step, the operator needed to label the item characteristics through a combo box-based interface composed of a hierarchical series (category, type, and material) of interdependent choices. For the aim, we slightly modified the ML shortlist proposed by the previous inter-operator reliability study [34]. The simplified list (Figure 2) included three main ML characteristics, namely type (characterised by their main potential source), material, and colour. Litter type also included the labels i) Fragments, pieces of an object with undefined shape and anthropogenic origin that could not be associated with any ML types present in the list, and ii) Undefined Items, objects that had a distinct shape but could not be associated with any type of ML present in the list, for being not visible and/or not recognisable enough due to operator inexperience and/or low image resolution. The *Undefined* option was also included in the material and colour lists, in case these characteristics could not be precisely defined by the operator. The tool also automatically retrieved marked item geometric properties (length, area, and coordinates). The colour property was an independent field instead.

Within the broader citizen science project "Adotta una spiaggia/Adopt a beach" (https://sites.google.com/view/seacleaner/educazione/adotta-una-spiaggia, accessed on 10 May 2021), 34 students from three different secondary school classrooms in Tuscany (Italy) were invited to perform the MS. Since all students had no previous experience in ML mapping and QGIS application, an online training course (four hours) was held to introduce the issue of ML in the coastal environment and the use of QGIS for performing the MS. Students also received supporting material comprising (i) a catalogue of ML images extracted from orthophotos of the study site, (ii) a quick guide and a video-recorded lesson about the use of QGIS for ML marking on orthophoto, and (iii) a recorded video lesson showing the UAV operation in the field and the photogrammetry framework. The material is available on the project website (see also Data Availability).

A dedicated GIS package was also provided to the students, hereafter citizen science operators (CSO). The package included the QGIS project (qgz format) with the user-friendly interface, the database (gpkg format), and the map of the Test Area (GEO-TIFF format). The hierarchical series of interdependent combo boxes made the selection of litter properties easier. In order to have information on the source of different kinds of objects, it was specified that recognisable material must be marked considering its original definition, avoiding, for example, selecting *Fragments* for broken drinking bottles.

After the training phase, each CSO autonomously marked and classified ML items using the ad hoc QGIS interface (Figure 3). CSO returned the single geopackage (gpkg file) containing the ML map and the corresponding attribute table. Therefore, the final dataset was composed by the layer on the QGIS project with i) the geometric properties of ML and ii) the characteristics (type, material, colour, and size) of each item.

After the drone flight, the standard in situ visual census (hereafter, VC) was performed by some of the authors of this paper, following the OSPAR protocol [17]. Items were categorised using the same criteria used for the MS (Figure 2), classifying items by dimensional class, type, colour, and material. The smallest size of items to be considered was set to 5 cm (instead of 2.5 cm) to have a better comparison between MS and VC. In fact, a previous study observed that the highest discrepancy between items collected in the field and identified on UAV images was in the range 2.5–5 cm [8]. To assess the reliability of CSO contributions in the marking phase, we compared the CSO output with the VC.

| Sources | id | Type |
|---|---|---|
| Fragments | 1 | Fragments |
| Containers | 2 | Bottles drinking (whole or fragments) |
| | 3 | Bottles cleaning products or similar items (whole or fragments) |
| | 4 | Caps or drums and buckets |
| | 5 | Bins, tanks, buckets, flower pots (whole or fragments) |
| | 6 | Tins/food cans or drink package drinks (whole or fragments) |
| | 7 | Tablewares, cups e plastic cups (whole or fragments) |
| | 8 | Forks, sticks, straws, chip foks, chopstick (whole or fragments) |
| | 9 | Caps and rink package rings |
| | 10 | Boxes and cardboards (whole or fragments) |
| | 11 | Aerosol cans |
| | 12 | Other containers (whole or fragments) |
| Fishing/Boating& Packaging material | 13 | Buoys (whole or fragments) |
| | 14 | Fishing nets, fishing line, fishing floats |
| | 15 | Crates (whole or fragments) |
| | 16 | Bags (whole o fragments) |
| | 17 | Mussel farming nets (whole or fragments) |
| | 18 | Packaging: ropes, strings, nets (whole or fragments) |
| | 19 | Other packaging items (whole or fragments) |
| | 20 | Nautical ropes |
| | 21 | Other fishing & boating items (whole or fragments) |
| Sanitary | 22 | Masks |
| | 23 | Cotton buds |
| | 24 | Gloves |
| | 25 | Syringe |
| | 26 | Medical vials (whole or fragments) |
| | 27 | Other medical items (whole o fragments) |
| Clothing | 28 | Shoes, flip-flops, boots, clogs (whole or fragments) |
| | 29 | Gloves (whole or fragments) |
| | 30 | Belt, bag, hat or other clothing, ecc. (identifiable) |
| | 31 | Other clothing not identifiable |
| Others | 32 | Light globes (whole or broken), light tubes |
| | 33 | Pallets /processed timbers (whole or fragments) |
| | 34 | Lighter |
| | 35 | Tyres |
| | 36 | Magazines, newspapers (whole or fragments) |
| | 37 | Construction items/ceramic items (whole or fragments) |
| | 38 | Household appliance |
| | 39 | Metallic nets |
| | 40 | Tubes |
| | 41 | Electrical wires |
| | 42 | Toys (ball, toy soldiers, toy fragments), (whole or fragments) |
| | 43 | Undefined |

| id | Detailed Materials |
|---|---|
| PL | Plastic |
| UD | Undefined |
| PS | Polystyrene |
| WO | Wood |
| TX | Textile (Leather, Fibers not synthetic) |
| CE | Ceramic |
| FO | Foam |
| GL | Glass |
| ME | Metal |
| MU | Multimaterial |
| PA | Paper |
| RU | Rubber |
| ST | Straw |
| TE | Tetrapack |

| id | Color |
|---|---|
| b | Blue |
| k | Black |
| y | Yellow |
| r | Red |
| gr | Green |
| n | Brown |
| v | Violet |
| g | Grey |
| w | White |
| t | Transparent |
| ud | Undefined |

| id | Size |
|---|---|
| S | Small |
| M | Medium |
| L | Large |

**Figure 2.** Classification of marine litter on UAV images. Left: macro-categories (source) and specific typology (Type) with the specific identity number (id). Types of litter are grouped into macro-categories to facilitate the choice during the marking process; right: materials of litter and their related id (upper), colour list, and related id (middle) and size of litter and specific id (lower). Item size: small between 5 and 15 cm; medium between 15 and 50 cm; large if bigger than 50 cm.

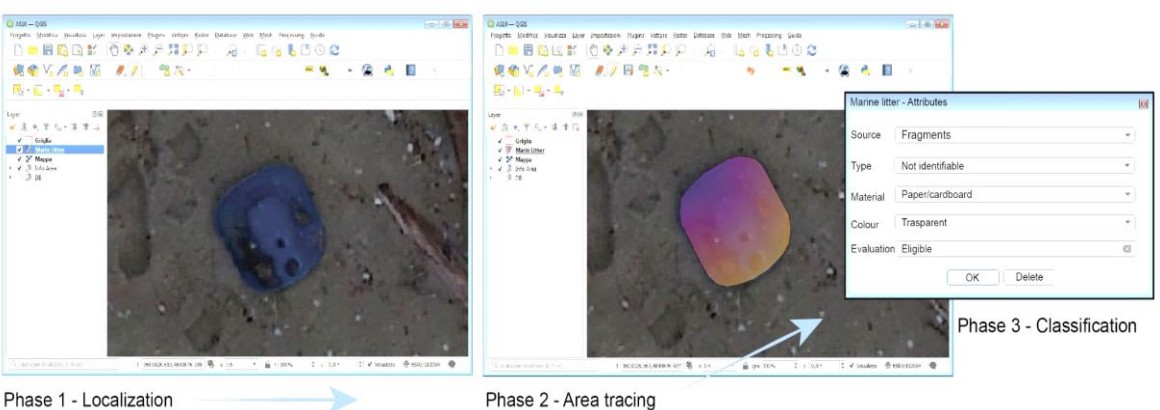

**Figure 3.** Screenshots of the developed software application for marine litter identification and classification with combo boxes.

### 2.3. Data Quality Assessment and Interoperator Concordance Test

A preliminary screening test was drawn to discard possible unreliable data. We considered a sub-sample of 42 objects particularly visible and of undoubted interpretation on the orthophoto (Figure 4). Firstly, we matched the correct objects marked by each CSO

with the truth layer, checking the classification (for each attribute of the object) with the classification in the truth layer. Second, we calculated the overall percentage of correct classifications made by each CSO against the total number of objects, setting a minimum acceptable threshold to 50%. Finally, we evaluated the degree of reliability of CSO in recognising, marking, and classifying the objects in the test area, discarding any CSO that did not meet the defined criteria.

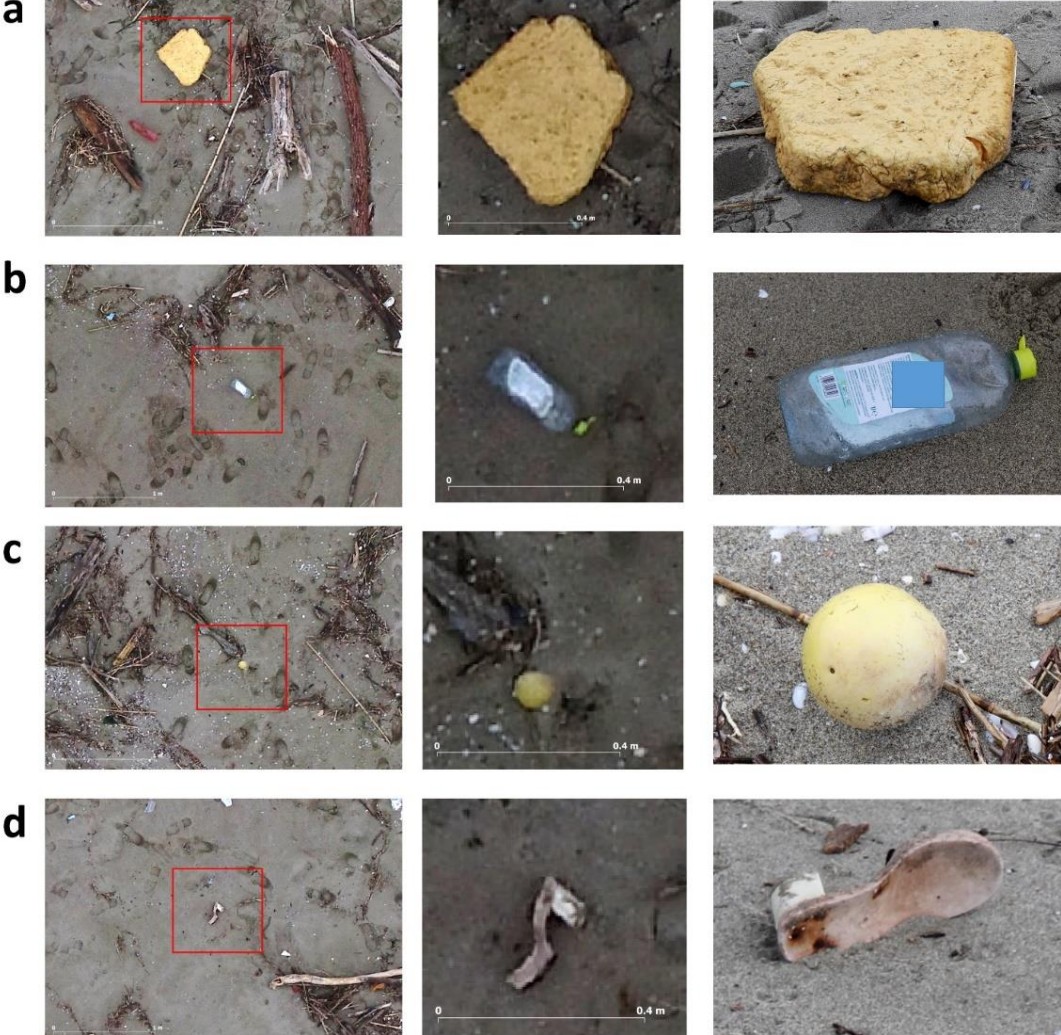

**Figure 4.** Examples of specific known items found on San Rossore beach. (**a**) Yellow foam fragment; (**b**) transparent plastic container; (**c**) yellow ball; (**d**) brown shoe. The left column shows the items (red square) on the orthophoto, the central column the items visible at 1:4 zoom factor on the orthophoto, and the right column the item picture taken in situ.

We also adopted the Kendall's coefficient of concordance (W) [53] to measure the level of agreement among the working groups in MS assessments. The test evaluated the level of concordance in detecting the number of ML items on the UAV image and in labelling the ML characteristics (type, material, colour, and dimension of items). Kendall's coefficient of concordance ranges from 0 (no agreement) to 1 (complete agreement).

### 2.4. Evaluation of Citizen Science Operators Assessments

Besides the comparison with the VC, a sub-sample of 100 items was randomly selected by an expert operator from among the ML bulk collected in the field. The CSO classification of the sub-sample was evaluated through three indicators, namely true positive rate (*TPR*), false positive rate (*FPR*), and positive predicted value (*PPV*) [54].

The true positive rate (*TPR*), also called sensitivity, measured the capacity of the operator in classifying item properties. It was computed as:

$$TPR = \frac{TP}{P} \times 100 \tag{1}$$

where *TP* was the number of correctly classified items considering one single property (e.g., items classified as plastic also found as plastic in the field), and *P* was the total number of real items within the same property (e.g., total number of plastic items in the Test Area from the field survey).

The false positive rate (*FPR*), also called the false alarm ratio, measured the probability of assigning a wrong property to an item.

$$FPR = \frac{FP}{N} \times 100 \tag{2}$$

where *FP* is the number of incorrectly classified items within one property (e.g., number of items classified as plastic that from field survey were another material), and *N* was the total number of real items that were not within one property (e.g., total number of items that were not plastic in the field).

*PPV* is the combination of the previous two and indicates the probability that an item has been classified correctly:

$$PPV = \frac{TPR}{(TPR + FPR)} \tag{3}$$

All three indicators varied between 0 and 100. Better assessment is indicated by a higher value of *TPR* and *PPV* and lower values of *FPR*.

## 3. Results

### 3.1. In Situ Visual Census and Manual Image Screening by Citizen Science Operator

Among the 34 citizen science operators (CSO) that completed the manual image screening (MS) task, just 30 works were considered for the evaluation. In fact, after the preliminary screening test, the data returned by two CSO were a copy of other colleagues.

The in situ visual census (VC) collected 332 ML items (Figure 5). Most of the ML bulk was composed of fragments (46%) and containers (30%). Fishing-related and other items were found in similar percentage (7%). Plastic composed about 56%, polystyrene 22%, while white was the most abundant colour (46%) (Figure 6).

On average, CSO marked about 49% of ML collected by in situ VC (Figures 5 and 6). Nevertheless, percentages of sources were fairly in agreement with the VC. Fragments were overestimated (53%), whereas fishing-related items (4%) and clothing (2.4%) were slightly underestimated. Overall, materials were not properly classified due to the fact that CSO were not able to identify about 22% of items. However, plastic and polystyrene composed more than the 50% of ML bulk from MS. This confirmed that the composition of ML items is a difficult property to identify from UAV images, especially by non-expert operators. As previously observed, white colour may mislead in the material classification.

Most of items collected in situ were small (86%). From MS, small items only composed 62%, whereas medium items 37% (Table 1).

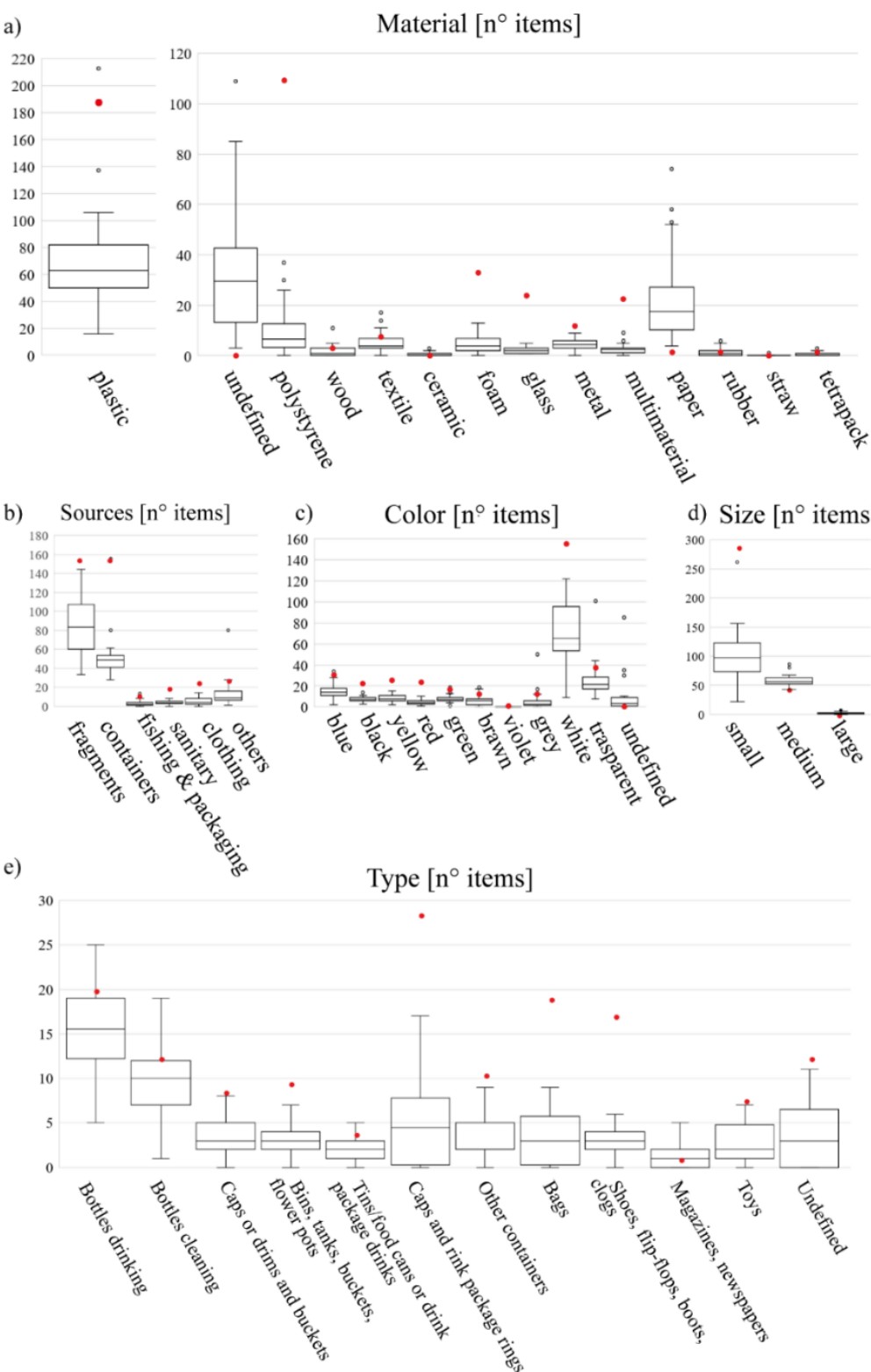

**Figure 5.** Boxplots of number of items marked by manual image screening (MS) and found by in situ visual census (VC, red dots), for material (**a**), source (**b**), colour (**c**), size (**d**), and type (**e**) categories. White dots show outliers. Item size: small between 5 and 15 cm; medium between 15 cm and 50 cm; large bigger than 50 cm.

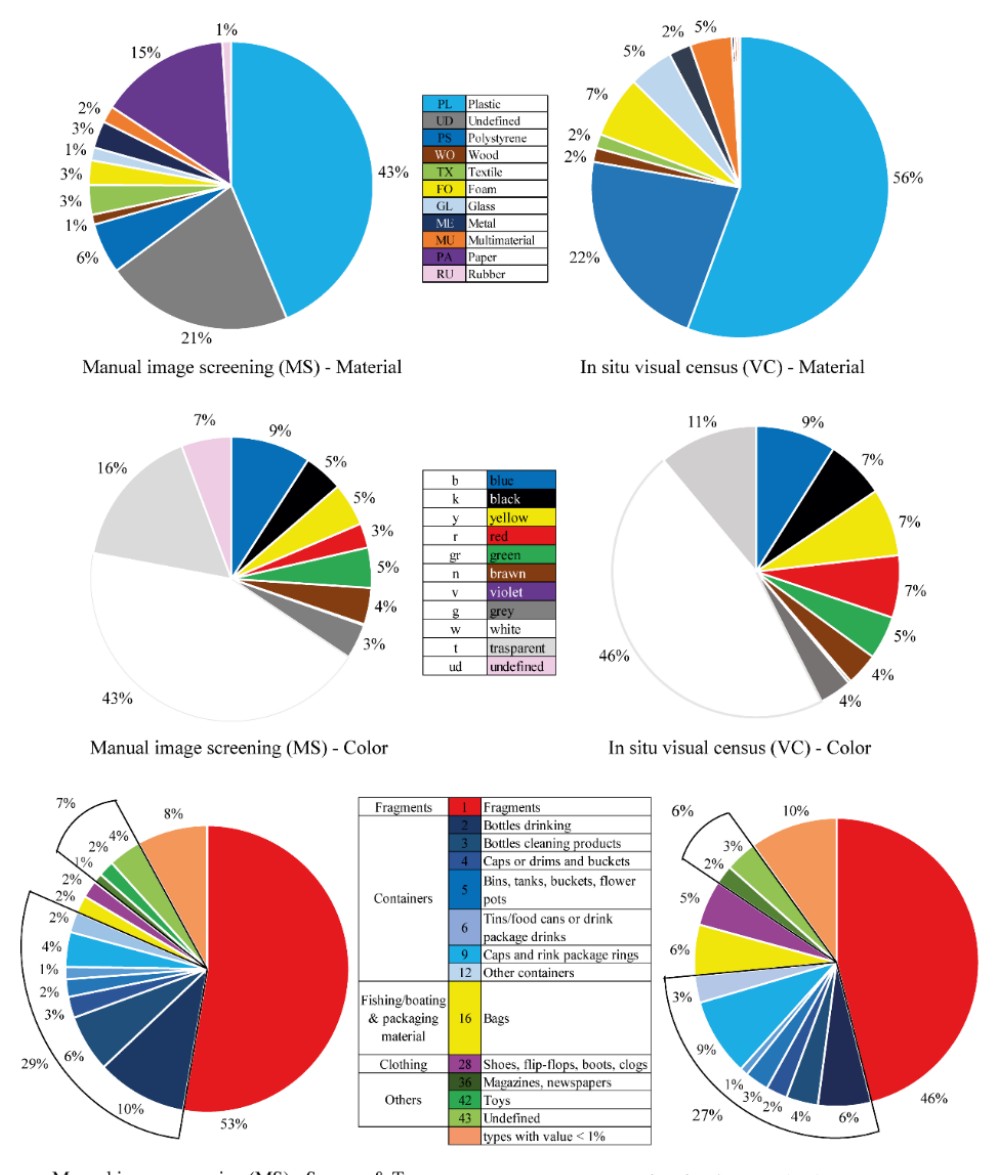

**Figure 6.** Percentages of items marked by manual image screening (MS) and collected during the in situ visual census (VC). Types not reaching 1% of the total were not individually graphed.

**Table 1.** Items classification based on size, collected in situ visual census (VC) and marked during the manual image screening (MS) by a citizen science operator (CSO).

| Size | VC (Number) | VC (%) | MS (Average Number) | MS (Average %) |
|---|---|---|---|---|
| Small (>5 cm) | 285 | 85.8 | 100 | 62.2 |
| Medium (15–50 cm) | 43 | 12.9 | 59 | 36.5 |
| Large (>50 cm) | 4 | 1.3 | 2 | 1.3 |

### 3.2. Citizen Science Operators Detection Performance

The Kendall W concordance value (Table 2) was lowest for the identification of the type (0.6) and highest for the size (W = 0.91). The results showed an inverse relationship between the number of options the operator has to choose and the concordance level. Nevertheless, achievements were similar to the agreement among experts of a previous

study [34], indicating that the skills of the CSO are low, dependent on their background and age. The agreement was comparable for all ML characteristics, namely type (0.60 vs. 0.58), material (0.75 vs. 0.76), and colour (0.69 vs. 0.65), suggesting that the interpretation of UAV images for an ML survey by CSO can also be robust, if they are properly trained.

**Table 2.** Kendall W Concordance test (W) among citizen science operators (CSO) for category, type, material, colour, and class size calculated based on the number of items.

| Attribute | Number of Categories | W |
|:---:|:---:|:---:|
| Type | 43 | 0.60 |
| Material | 14 | 0.71 |
| Colour | 11 | 0.69 |
| Size | 3 | 0.91 |
| Source | 6 | 0.86 |

The ability of the CSO group to correctly recognise ML objects, evaluated on a sample of 100 objects randomly selected by experts, returned satisfactory results (Figure 7). On average, CSO presented high scores in the classification of the type, colour, and size (*PPV* = 94%), whereas achievements were slightly worse for material (*PPV* = 76%) categorisation. CSO obtained high scores in the classification of the type, colour, and size (*PPV* = 94%), whereas achievements were slightly worse for material (*PPV* = 76%).

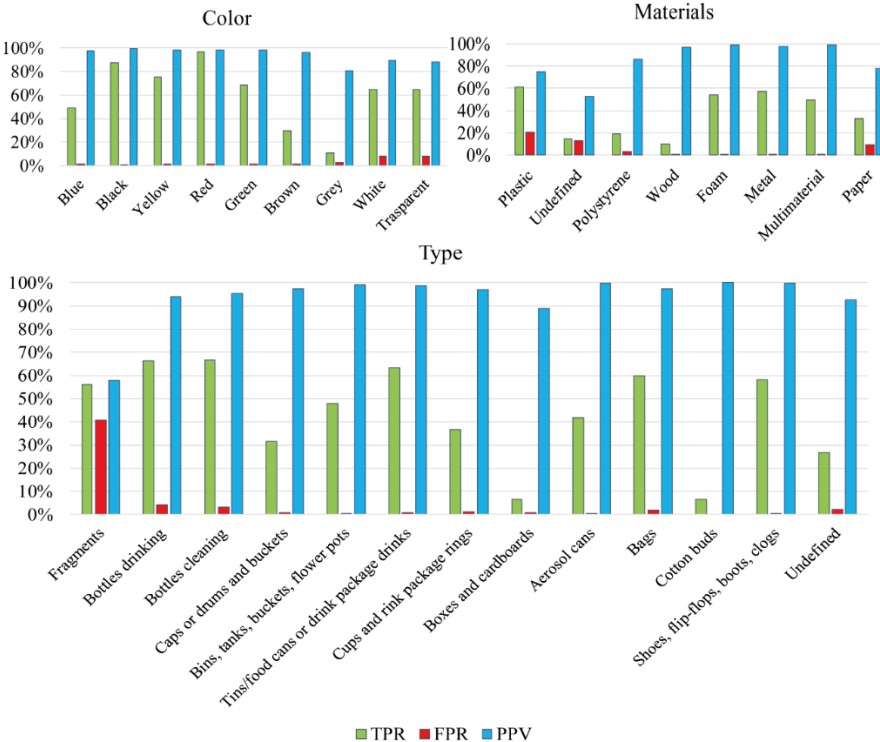

**Figure 7.** Citizen science operators (CSO) classification for the subsample of 100 items. Average *TPR* (green bars), *FPR* (red), and *PPV* (blue) for colour (upper left), material (upper right), and type (lower).

Most of the CSO identified more than 50% of the ML, obtaining the highest score for toys (*TPR* = 73%) and more than 60% for containers. On the other hand, CSO had difficulties in the identification of cardboard and cotton buds (*TPR* < 10%). Fragments were shown to be the most difficult to classify (*FPR* = 40% and *PPV* = 58%). Nevertheless, *PPV* were highly variable for those items mostly found as fragments (buckets, containers, tires, and toys), whereas final *PPV* was most homogeneous for common items such as bottles, cans, and shoes (Figure 7).

Regarding the ML materials (Figure 7), the highest *TPR* was scored for plastic (60%), and the lowest was for wood (10%). Due to the high values of *FPR*, the final *PPV* was low for undefined material (53%), plastic (75%), and paper (78%), whereas all other materials were properly identified by CSO (96%, on average). The low results in classifying plastics and papers may be because 60% of white plastic items were erroneously classified as paper items; thus, the white colour misled the CSO.

Regarding ML colour classification (Figure 7), *TPR* was highest for primary colours (e.g., Red *TPR* = 97%), while it was lowest for those chroma that had less contrast with the beach background and pieces of natural wood, such as grey (11%) and brown (29%), respectively. Although grey had the highest *PPV* (82%), most grey items (53%) were wrongly labelled as white. Similarly, 30% of brown items were labelled as yellow. These inaccuracies may be due to the high subjectivity in choosing between colours that look similar in the UAV aerial images.

Since the provided user-friendly application helped CSO in digitising the items' contour, the size classification returned a high *TPR* for both medium and large items (93%). The lowest *TPR* and highest *FPR* were instead returned for small items, for a final *PPV* of 87%. On average, most of the CSO (90%) classified ML size with a *PPV* higher than 80%.

## 4. Discussion

This study evaluated the reliability of non-expert citizen scientist operators (CSO) (students of secondary schools) in marking and classifying marine litter (ML) items from aerial photos taken by an unmanned aerial vehicle (UAV, aka drone). Overall, results confirmed that citizen science projects can support UAV-based ML survey upon a proper training program and the provision of a user-friendly guided software. In fact, it was fundamental to introduce CSO to ML issues and provide (i) video tutorials, (ii) visual manual instructions regarding manual image screening (MS) for item detection on UAV images, and (iii) a QGIS application built for guiding the operator during MS.

Comparing CSO assessments with those of a previous UAV-based survey performed by experts [8] in the same geographical area, we found that the percentage of material correctly classified was slightly better for plastic (38% vs. 20%) and worse for metal (59% vs. 66%) and glass (19% vs. 66%). Yet, the classification for dimensional class (size) was more satisfactory for smaller items (35% against 20%). Instead, CSO overestimated the number of medium (36%) and large items (56%), while experts correctly marked 90% and 70% of these size categories [8].

The operational training sessions to identify a limited and predefined set of items present in the study area, followed by the analysis of the errors made by CSO in marking and classifying specific litter items, might be a way to improve a priori knowledge of the CSO. A priori CSO knowledge of some variables might be wrong, and therefore bias could be introduced. For instance, CSO overcounted paper items from images, underestimating instead polystyrene white pieces (probably misleading white items according to their prior knowledge about the state of "tourist" beaches), whereas experts were aware that paper is not present at the study site; the area is not accessible to tourists, away from sources of anthropogenic material, and previous surveys did not encounter this material [8,40,41]. This information was deliberately not given to CSO to avoid influencing their choice during the MS. A white object with irregular edges can easily be mistaken for a piece of paper if one does not have prior knowledge that such a specific object is practically absent in certain conditions. The a priori knowledge gap of non-experts can be therefore partly corrected by specific training sessions.

On the other hand, the ability of CSO to correctly assign the source of identified ML, and to recognise, for many of them, the type, was confirmed. The most significant discrepancies between the two surveys are both the higher percentage of 'Undefined' items marked by MS compared to VC and the lower number of localised objects in the ortho-photos, particularly small ones.

Regarding the Data Availability on the classification of 100 items, positive predictive value (*PPV*) ranged between 76% and 94% in the classification of type and material from UAV images, a good quality assessment considering that this was the first experience in ML study, UAV image processing, and QGIS operation and that the CSO involved had no experience of either marine litter or mapping with QGIS.

The Kendall (W) level of agreement among CSO both in the identification and classification of ML on drone images was similar to the agreement among experts [34], indicating that the skills of the CSO are low dependent on their background and age. The agreement was comparable for all ML characteristics, therefore the interpretation of UAV images for ML survey can also be robust by CSO, if properly trained.

Results obtained from the present study, carried out on the test area, confirm that the citizen science program greatly increases the possibility of obtaining reliable data over large areas and long periods and can be used for the spatial and temporal ML distribution through UAV orthophoto.

We underline that a preliminary screening of data assessments to discard low-quality MS and/or uncompleted works was necessary. The attention and effort given during the MS by each operator is a difficult factor to weigh, as already pointed out by Andriolo et al. [34] for experts. We reiterate this concept, specifying that the user-friendly application helped and facilitated CSO during the alienating and tedious MS task, limiting the time spent in marking and decreasing fatigue. This fact was found in the particular case of two students, who used the work done by others, thus not producing their own personal results. In cases such as this, extra care must be taken when selecting data, compared to the case of non-expert citizens who volunteered for this type of activity. Being a CSO group composed of students, some lack of willingness to participate was expected; however, most of them returned good quality data.

Besides the MS task, future citizen science programs may also propose the involvement of citizens in the aerial image acquisition [55]. The actual diffusion of low-cost drones may advance the collection of stranded litter images, helping in improving the spatial and temporal coverage of coastal pollution.

## 5. Conclusions

This study presented a citizen science program that involved students in detecting and mapping marine litter (ML) on unmanned aerial vehicle (UAV) images. A specific framework was built for training the students, named citizen science operators (CSO) here. A CSO training framework included an introduction to the ML issue and a practical session on the use of QGIS application. The framework was shown to be efficient and may be useful to implement citizen science projects.

The comparison with the results of the in situ visual census (VC) showed an underestimation of the items number, with only 49% marked on the image. In particular, a large fraction of small size items were not recognised on UAV images. Nevertheless, the sources of the ML were properly identified, and, overall, the ML bulk was properly described in terms of percentage. The difficulty in correctly defining ML materials from UAV images was confirmed in this work. Additionally, it was observed that knowledge of the most common items in the area is critical. For this reason, the training phase should include a session dedicated to the characterisation of ML previously found in the area.

As the assessment accuracy was in line with previous works developed by experts, remote detection of marine litter on UAV images can be improved through citizen science programs, upon an appropriate training plan and provision of specific tools.

**Author Contributions:** Conceptualisation, S.M. and L.M.; methodology, S.M., M.P., L.M., U.A. and G.G; software, M.P. and L.M.; validation, S.M., M.L., M.P., L.M., U.A. and G.G.; formal analysis, L.M.; investigation, S.M., M.L., M.P. and L.M.; resources, S.M., M.L., M.P. and L.M; data curation, S.M., M.P. and L.M.; writing—original draft preparation, S.M. and L.M.; writing—review and editing, S.M., M.L., M.P., L.M., U.A. and G.G.; visualisation, M.L., M.P. and L.M.; supervision, S.M.; project administration, S.M.; funding acquisition, S.M. and M.P. All authors have read and agreed to the published version of the manuscript.

**Funding:** This paper is part of NAUTILOS project that has received funding from the European Union's Horizon 2020 research and innovation programme under grant agreement No. 101000825. This work was supported by the Portuguese Foundation for Science and Technology (FCT) and by the European Regional Development Fund (FEDER) through COMPETE 2020, Operational Program for Competitiveness and Internationalization (POCI) in the framework of UIDB/ 00308/2020 and the research project UAS4Litter (PTDC/EAM-REM/30324/2017).

**Institutional Review Board Statement:** Not applicable.

**Informed Consent Statement:** Not applicable.

**Data Availability Statement:** Datasets generated during this study, and material supporting the training phase of CSO can be found in https://sites.google.com/view/seacleaner/educazione/adotta-una-spiaggia, accessed on 4 August 2021.

**Acknowledgments:** We would like to thank the Park of Migliarino, Massacciuccoli, and San Rossore for the permission to access the protected area and perform the field experience. Thanks to all the co-authors for the commitment to participating in this work and for providing the high-quality data necessary to perform the inter-operator concordance test. A special thanks goes to the involved scholastic institutes, to the students that participated in this citizen science and educational experience during the hard days of COVID emergencies, and to their teachers that supported them and helped us in collecting data: for IIS Meucci of Massa (MS), Fabio Pieraccioni; for IIS Zaccagna-Galilei of Carrara (MS), Chiara Collotti and M. Cristina Matelli; for IIS Agnoletti of Sesto Fiorentino (FI), Laura Dei; for ISI Garfagnana of Castelnuovo (LU), Andrea Malagoli; for LS Marconi San Miniato (PI), Laura Doria.

**Conflicts of Interest:** The authors declare no conflict of interest.

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
