# Peer review of "Citizen Science for Marine Litter Detection and Classification on Unmanned Aerial Vehicle Images"

_water, doi:10.3390/w13233349_

Round 1

Reviewer 1 Report

The paper entitled “Citizen Science for Marine Litter Detection and Classification on Unmanned Aerial Vehicle Images” is a sound and well-written paper, referring to a very important current issue, namely citizen science projects.

Since citizen science issues are more and more discussed lately, it should be promoted accordingly where conditions are suitable. I deeply believe that the present manuscript will provide essential assistance towards this direction and therefore I highly recommend it for publication after minor (mainly linguistic) revision (see attached file).

Author Response

Our answers in the attached file.

Many thanks

Reviewer 2 Report

This is a trial to evaluate the reliability of  non-expert citizen scientist in marking and classifying marine litter. Considering the future framework of citizen scientist participation in practical issues, this kind of trial have a significance. 

Although I do not know the topic, i.e. application of drones finding marine litter, is interesting for readers or not, the framework of the comparison on the results by citizen scientist and specialist is interesting, and beneficial for the society to give a clue on similar type of studies.

The reviewer has some small questions, and adding appropriate explanations on these questions in the paper is preferable.

(1) The school system are different in other countries. So, readers cannot imagine the age of the 34 students.

(2) In Figure 4, there are no indexes of a) to d). Maybe the top line is a), and the bottom line is d), but it should be indicated.

(3) In figure 5 e), I do not feel it is better to show numbers on the indexes. I.e., 2-Buttles drinking, 3-bottles cleaning, etc.,...I wondered what is 2 and 3?

(4) In Figure 5, I can find the red dots as the results of in-situ visual census. But what are these black dots? (black small circles?)

(5) In Figure 6, the subtitles of each plots shall be the bottom of the corresponding plots, I suppose. Not at the top.

Author Response

(The authors gave the same response as above.)
